# Afucosylated *Plasmodium falciparum*-specific IgG is induced by infection but not by subunit vaccination

Mads Delbo Larsen [1,2,12], Mary Lopez-Perez [3,12], Emmanuel Kakra Dickson[4], Paulina Ampomah[5], Nicaise Tuikue Ndam[6], Jan Nouta[7], Carolien A. M. Koeleman[7], Agnes L. Hipgrave Ederveen [7], Benjamin Mordmüller[8,9], Ali Salanti [3], Morten Agertoug Nielsen [3], Achille Massougbodji[10], C. Ellen van der Schoot[1,2], Michael F. Ofori[4], Manfred Wuhrer [7], Lars Hviid [3,11,13✉] & Gestur Vidarsson [1,2,13✉]

*Plasmodium falciparum* erythrocyte membrane protein 1 (PfEMP1) family members mediate receptor- and tissue-specific sequestration of infected erythrocytes (IEs) in malaria. Antibody responses are a central component of naturally acquired malaria immunity. PfEMP1-specific IgG likely protects by inhibiting IE sequestration and through IgG-Fc Receptor (FcγR) mediated phagocytosis and killing of antibody-opsonized IEs. The affinity of afucosylated IgG to FcγRIIIa is up to 40-fold higher than fucosylated IgG, resulting in enhanced antibody-dependent cellular cytotoxicity. Most IgG in plasma is fully fucosylated, but afucosylated IgG is elicited in response to enveloped viruses and to paternal alloantigens during pregnancy. Here we show that naturally acquired PfEMP1-specific IgG is strongly afucosylated in a stable and exposure-dependent manner, and efficiently induces FcγRIIIa-dependent natural killer (NK) cell degranulation. In contrast, immunization with a subunit PfEMP1 (VAR2CSA) vaccine results in fully fucosylated specific IgG. These results have implications for understanding protective natural- and vaccine-induced immunity to malaria.

[1] Department of Experimental Immunohematology, Sanquin Research, Amsterdam, The Netherlands. [2] Landsteiner Laboratory, Amsterdam UMC, University of Amsterdam, Amsterdam, The Netherlands. [3] Centre for Medical Parasitology, Department of Immunology and Microbiology, Faculty of Health and Medical Sciences, University of Copenhagen, Copenhagen, Denmark. [4] Department of Immunology, Noguchi Memorial Institute for Medical Research, University of Ghana, Accra, Ghana. [5] Department of Biomedical Sciences, School of Allied Health Sciences, University of Cape Coast, Cape Coast, Ghana. [6] Université de Paris, MERIT, IRD, 75006 Paris, France. [7] Center for Proteomics and Metabolomics, Leiden University Medical Center, Leiden, The Netherlands. [8] Department of Medical Microbiology, Radboud University Medical Center, Nijmegen, The Netherlands. [9] Institut für Tropenmedizin, Universitätsklinikum Tübingen, Tübingen, Germany. [10] Centre d'Etude et de Recherche sur le Paludisme Associé à la Grossesse et à l'Enfance (CERPAGE), Faculté des Sciences de la Santé, Université d'Abomey-Calavi, Godomey, Benin. [11] Centre for Medical Parasitology, Department of Infectious Diseases, Rigshospitalet, Copenhagen, Denmark. [12] These authors contributed equally: Mads Delbo Larsen, Mary Lopez-Perez. [13] These authors jointly supervised this work: Gestur Vidarsson, Lars Hviid. ✉email: lhviid@sund.ku.dk; g.vidarsson@sanquin.nl

The most severe form of malaria is caused by the protozoan parasite *Plasmodium falciparum*. The disease is currently estimated to cost around 400,000 lives a year, mostly of young children and pregnant women in sub-Saharan Africa. In addition, nearly 900,000 babies are born with low birth weight as a consequence of placental malaria (PM)[1]. The particular virulence of *P. falciparum* is related to the efficient adhesion of the infected erythrocytes (IEs) to host receptors in the vasculature, such as endothelial protein C receptor, intercellular adhesion molecule 1, and oncofetal chondroitin sulfate A[2–4], mediated by members of the protein family *P. falciparum* erythrocyte membrane protein 1 (PfEMP1), embedded in the membrane of IEs[5]. The sequestration of IEs can cause tissue-specific circulatory compromise and inflammation, which in turn can lead to severe and life-threatening complications such as cerebral malaria (CM) and PM[6,7]. Severe malaria in children has repeatedly been shown to be associated with parasites expressing particular subsets of PfEMP1, such as Group A and B/A[4,8], whereas PM is strongly associated with parasites expressing VAR2CSA-type PfEMP1 (refs. [9,10]).

Acquired protective immunity to *P. falciparum* malaria is mainly mediated by IgG with specificity for antigens expressed by the asexual blood-stage parasites[11]. PfEMP1 is a key target[5], although antibodies to other blood-stage antigens such as the merozoite-specific antigens glutamate-rich protein (GLURP), merozoite surface protein 1, and reticulocyte-binding protein homolog 5 also contribute to naturally acquired protection[12–14]. Importantly, the selective protection from severe malaria that develops early in childhood is related to acquisition of IgG specific for Group A and B/A PfEMP1 variants[8,15,16]. As a result, life-threatening complications are rare in teenagers and beyond in *P. falciparum* endemic regions. PM, which is caused by selective accumulation of VAR2CSA-positive IEs in the placenta from early in pregnancy[17,18], constitutes an important exception to this rule. Only VAR2CSA mediates adhesion to placenta-specific chondroitin sulfate[9,19]. Because of this, and because antibodies specific for non-pregnancy-related types of PfEMP1 do not cross-react with VAR2CSA[9,20], primigravid women are immunologically naïve to VAR2CSA and therefore highly susceptible to PM despite general protective immunity acquired during childhood. However, substantial IgG-mediated protection against PM is acquired in a parity-dependent manner, and PM is therefore mainly a problem in the first pregnancy[9,21–23].

Acquired immunity mediated by PfEMP1-specific IgG is generally thought to rely on its ability to interfere directly with IE sequestration (i.e., neutralizing, adhesion-inhibitory antibodies). However, antibody-mediated opsonization of IEs is a likely additional effector function of these antibodies, since the antibody response to most *P. falciparum* asexual blood-stage antigens (including PfEMP1) is completely dominated by the cytophilic subclasses IgG1 and (to a lesser extent) IgG3 (refs. [24,25]). Nevertheless, the relative importance of neutralization and opsonization remains largely unexplored. Complement-mediated destruction of IgG-coated IEs does not seem important[26], suggesting that IgG opsonization of IEs by IgG functions mainly through IgG-Fc receptor (FcγR)-dependent phagocytosis and antibody-dependent cellular cytotoxicity (ADCC)[27–29]. The latter involves FcγRIIIa[30]. Binding of IgG to FcγRIIIa critically depends on the composition of a highly conserved N-linked glycan at position 297 in the Fc region[31]. Different monosaccharides, such as galactose, fucose, and sialic acids, are added to the bi-antennary core structure of the Fc glycan (Fig. 1c). The level of fucosylation is of particular significance, since afucosylated IgG has up to 20-fold increased affinity for FcγRIIIa[32]. The reason is believed to be a steric clash between a unique N-linked glycan at position N162, not found in FcγRIa nor FcγRIIa/b, and the bi-antennary glycan

in the IgG-Fc imposed by the core-fucose[32]. Even more strikingly, IgG-afucosylation can convert a non-functional ADCC potential to strong and clinically significant responses[33–36]. Increased galactosylation at N297 can further enhance affinity to FcγRIII by additional twofold, and also increases the complement activating capacity of the antibody. In contrast, no influence of bisecting *N*-acetylglucosamine (GlcNAc) on antibody effector functions has been demonstrated so far[35].

Fc fucosylation of plasma IgG is near 100% at birth, and although it decreases slightly with age, it normally remains high (~94%) in adults[37,38]. Nevertheless, very marked and clinically significant reductions (down to ~10%) in antigen-specific IgG-Fc fucosylation are frequently observed after alloimmunization against erythrocyte and platelet alloantigens[39–41]. Afucosylation has also been observed for antigen-specific IgG to various enveloped viruses[33,34,42]. In human immunodeficiency virus (HIV) infections, low Fc fucosylation has been proposed as a trait of elite controllers[42], but it is associated with FcγRIIIa-mediated immunopathology in SARS-CoV-2 and secondary dengue virus infections[33,34,43]. Vaccination with the attenuated paramyxoviruses measles and mumps also results in specific IgG with reduced fucosylation similar to that acquired after natural infection[34]. In contrast, infection with the non-enveloped parvovirus B19, protein subunit vaccination against hepatitis B virus, vaccination with inactivated influenza virus, or vaccination against tetanus, pneumococcal, and meningococcal disease do not induce selectively afucosylated IgG[34,44,45].

The above findings have led us to propose that afucosylated IgG has evolved as a beneficiary immune response to foreign antigens expressed on host membranes in the context of infections, which is mimicked in alloimmunizations with devastating consequences[34,40,41,46,47].

In this study, we show that antibody responses to *P. falciparum* antigens expressed on the IE surface are also a subject to afucosylation. Specifically, we show this for naturally acquired IgG responses to the PfEMP1 antigens VAR6 and VAR2CSA as well as its absences in responses to the merozoite antigen GLURP and VAR2CSA-specific IgG induced by subunit vaccination.

## Results and discussion

**Naturally acquired PfEMP1-specific IgG is highly afucosylated.** We first used a set of plasma samples collected from 127 pregnant Ghanaian women at the time of their first visit to antenatal clinics[48] to assess N297 glycosylation of IgG with specificity for three *P. falciparum* recombinant antigens. We used the full ectodomains of VAR2CSA and the non-pregnancy-restricted Group A-type VAR6, which are both naturally expressed on the IE surface. We also included the merozoite antigen GLURP, which is not expressed on IE surface[49] (Fig. 1).

In line with our hypothesis suggesting that afucosylated IgG response is restricted to foreign antigens expressed on host cells (such as alloantigens and outer-membrane proteins of enveloped viruses[34,40,41,46,47]), IgG1 responses to VAR6 and VAR2CSA were markedly Fc afucosylated (Fig. 2a). All individuals showed lowered anti-VAR6 Fc fucosylation compared to total IgG1, which remained high. The magnitude of the decreased Fc fucosylation of VAR6-specific IgG1 exceeded any previously reported pathogen-derived immune response. The most similar responses are against rhesus D on red blood cells and human platelet antigen-1a on platelets. However, IgG1 responses to those antigens display big variation in Fc fucosylation ranging from almost 100 to 10%[40,46]. In contrast, GLURP-specific IgG1-Fc fucosylation was generally high, also in line with our hypothesis (Fig. 2a). A few women showed marked afucosylation of GLURP-specific IgG1 (Fig. 2a), possibly in response to GLURP deposited

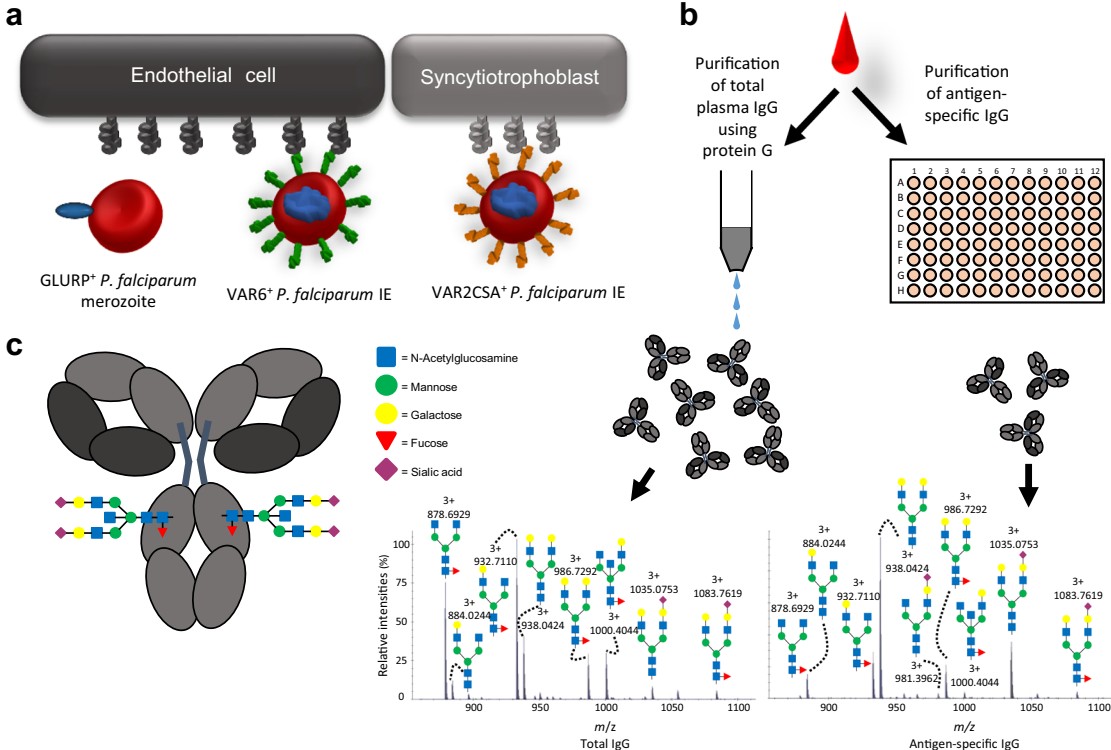

**Fig. 1 Background and study workflow. a** IgG1 specific for the merozoite antigen GLURP and two members of the PfEMP1 family expressed on the surface of IEs were analyzed in this study. Most PfEMP1 variants facilitate sequestration of IEs to vascular endothelium (exemplified here by VAR6), while VAR2CSA-type PfEMP1 mediate IE sequestration in the placental syncytiotrophoblast and intervillous space. **b** Plasma samples were split and used to purify total plasma IgG1 and antigen-specific IgG1, using protein G-coupled sepharose and solid-phase absorption with recombinant antigens, respectively. Eluted IgG1 was digested with trypsin and the glycopeptides analyzed by liquid chromatography mass spectrometry (LC-MS). Examples of MS spectra of total IgG1 (left) and antigen-specific (anti-VAR6) IgG1 (right) from one sample is shown. **c** The fractions of the different glycosylation traits of the Fc glycan depicted were calculated from LC-MS spectra.

on the erythrocyte surface during invasion, as has been described for other merozoite-specific antigens[50]. IgG1 specific for all three *P. falciparum* antigens showed higher Fc galactosylation and sialylation levels than total IgG1, similar to what is known for recent immunizations[34,44] (Supplementary Fig. 1A, B). Levels of bisecting GlcNAc were lower for VAR2CSA- and VAR6-specific IgG1, and higher for GLURP-specific IgG1 compared to total IgG1 (Supplementary Fig. 1C). These results indicate that antigen-specific IgG levels are modulated in complex ways according to exposure and antigen context.

Afucosylation of VAR2CSA-specific IgG1 was generally less pronounced than that of VAR6-specific IgG1 (Fig. 2a). Exposure to VAR2CSA-type PfEMP1 occurs later in life, as it is restricted to pregnancy, whereas *P. falciparum* expressing Group A PfEMP1 (such as VAR6) are associated with severe malaria in children[4,8]. IgG responses to Group A PfEMP1 variants are acquired from early in life in endemic areas through repeated exposure to parasites expressing those variants[15,16,51,52]. VAR6-specific IgG1 was consistently afucosylated in all tested individuals, probably as a result of continuous exposure to Group A PfEMP1 in childhood (Fig. 2a), suggesting that afucosylation is a persistent phenotype once acquired. This is further supported by the fact that VAR6-specific IgG-Fc fucosylation was negatively correlated with age, as was total IgG-Fc fucosylation similar to previous reports[37] (Supplementary Fig. 2). In contrast, the level of fucosylation of VAR2CSA-specific IgG1 was more varied (Fig. 2a) and decreased with increased antigen exposure, using parity as proxy (Fig. 2b). This was not the case for VAR6- (Fig. 2c) or GLURP-specific IgG1 (Fig. 2d), and only marginally so for total plasma IgG1 (Fig. 2e). VAR2CSA-specific IgG Fc fucosylation was also

negatively correlated with age (Supplementary Fig. 2A), most likely reflecting the strong correlation between parity and age. In contrast, no correlation was found between age and GLURP-specific IgG Fc fucosylation (Supplementary Fig. 2). The variability in afucosylation in the first pregnancy to VAR2CSA is in line with the stochastic afucosylation in primary alloimmune responses to red blood cell- and platelet antigens in pregnancies[40,47] as well as the stochastic levels of afucosylation in responses against different antigens within an individual[34]. Together, these results suggest that despite the initial variability, afucosylated IgG to PfEMP1 accumulates with repeated exposure.

No correlations between IgG levels and Fc fucosylation levels were observed (Supplementary Fig. 3). VAR2CSA-specific Fc fucosylation was not significantly different between women with or without PM ($P = 0.12$; Mann–Whitney test, Supplementary Fig. 4), and VAR2CSA-specific IgG-Fc fucosylation levels did not add significant predictive strength to generalized linear models predicting birth weight ($P = 0.071$) or maternal hemoglobin levels at delivery ($P = 0.19$) (Supplementary Tables 1 and 2). However, the striking pattern of decreasing Fc fucosylation with increasing parity provides a highly plausible molecular explanation of the well-known protective effect of multiple pregnancies towards PM. Studies aimed to establish the protective effect of afucosylated VAR2CSA-specific IgG in PM should be carried out in the future.

**Fc afucosylation of PfEMP1-specific IgG is stable**. The above findings support the hypothesis that afucosylated IgG specific for host membrane-associated immunogens is attained following repeated exposure and that the phenotype is stable once acquired. To examine this hypothesis further, and to consolidate the

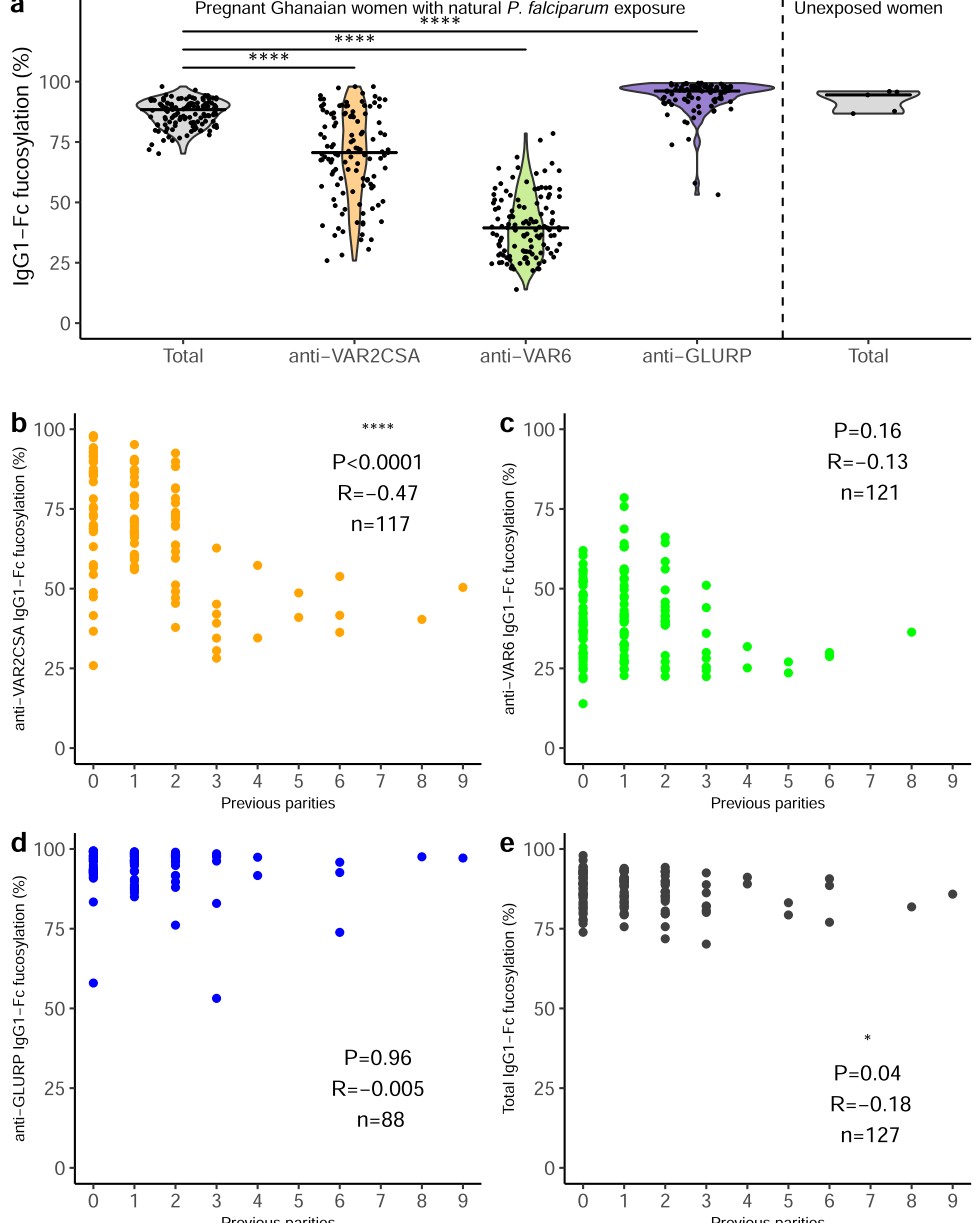

**Fig. 2 Fc fucosylation of naturally acquired *P. falciparum*-specific IgG depends on antigen location and exposure. a** Fc fucosylation levels of total plasma IgG1 (gray, *n* = 127) and IgG1 specific for VAR2CSA (orange, *n* = 117), VAR6 (green, *n* = 121), and GLURP (blue, *n* = 88) in Ghanaian pregnant women (left four panels). Fc fucosylation levels of total plasma IgG1 from unexposed Dutch women (*n* = 5) were included for comparison (right panel). Medians and densities are shown. Statistically significant pairwise differences between antigen-specific IgG and total IgG (multiple two-sided Wilcoxon signed-rank tests with Bonferroni correction) are indicated (****P* < 0.0001). **b**–**e** Correlations of **b** VAR2CSA-, **c** VAR6-, **d** GLURP-specific, and **e** total IgG1-Fc fucosylation levels with parity. *P* values and correlation coefficients are shown. Statistical significance of correlations (Spearman's correlations. **P* < 0.05; ***P* < 0.01; ****P* < 0.001; *****P* < 0.0001.

findings described above, we proceeded to determine the Fc fucosylation of IgG with specificity for the same three antigens, using an availability-based subset (*N* = 72) of plasma samples from a previously published cohort of Ghanaian women sampled while not pregnant[53]. The findings regarding total and antigen-specific IgG1 (Fig. 3 and Supplementary Fig. 1D–F) were fully consistent with those obtained with the samples from pregnant women. The marked Fc afucosylation of VAR2CSA- and VAR6-specific IgG1 was more pronounced among this second group of women (Fig. 3a and Supplementary Fig. 5), probably reflecting the more intense parasite transmission in the rainforest compared to the coastal savannah where the non-pregnant and pregnant

women were recruited, respectively[48,53]. In this cohort, we were also able to purify low level of VAR2CSA-specific IgG from nulligravidae, plausibly caused by unrecorded/undiscovered pregnancies. Although VAR2CSA-specific IgG levels decay markedly within 6 months of delivery[54,55], the parity-dependency of the degree of VAR2CSA-specific IgG1-Fc afucosylation remained in these non-pregnant women (Fig. 3b). Furthermore, there was no significant correlation between the time since last pregnancy and Fc fucosylation levels of VAR2CSA-specific IgG1 (Fig. 3c). Taken together, these findings reinforce the inference that PfEMP1-specific IgG1-Fc afucosylation remains stable in the absence of exposure to antigen. This conclusion is in line with our

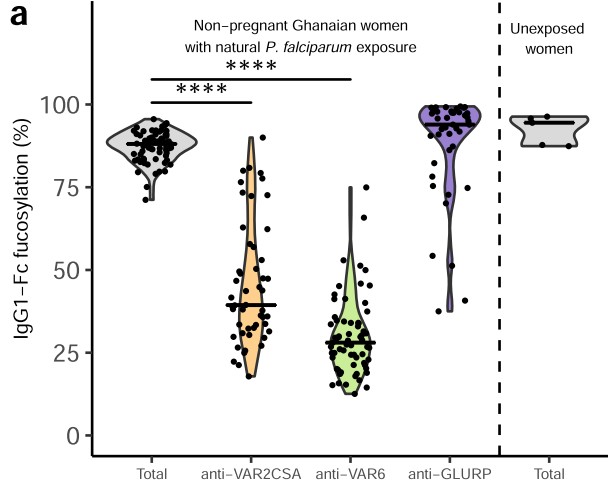

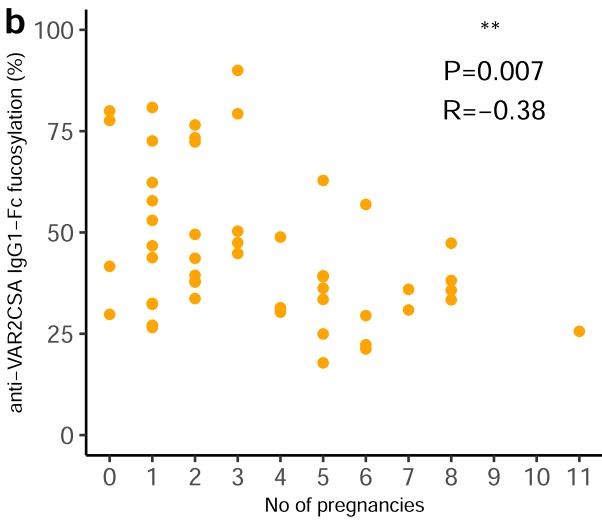

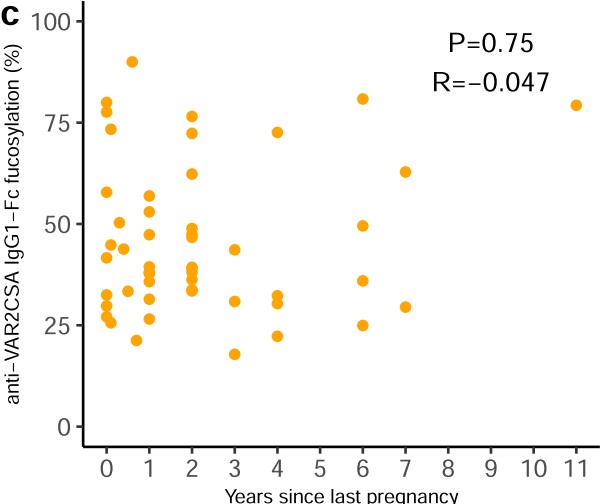

**Fig. 3 Fc fucosylation levels of VAR2CSA-specifc IgG is temporally stable. a** Fc fucosylation levels of total plasma IgG1 (gray, $n = 72$) and IgG1 with specificity for VAR2CSA (orange, $n = 50$), VAR6 (green, $n = 65$), and GLURP (blue, $n = 43$) in non-pregnant Ghanaian women exposed to VAR2CSA during one or more previous pregnancies. Fc fucosylation levels of total plasma IgG1 from unexposed Dutch females ($n = 5$) are included as controls. Medians and densities are shown. Statistically significant pairwise differences between antigen-specific IgG and total IgG (multiple two-sided Wilcoxon signed-rank tests with Bonferroni correction) are indicated (****$P < 0.0001$). **b** Correlation between fucosylation levels of VAR2CSA-specific IgG1 and parity. **c** Correlation between fucosylation levels of VAR2CSA-specific IgG1 and time since last pregnancy. Statistical significance of correlations are shown (Spearman's correlations. *$P < 0.05$; **$P < 0.01$; ***$P < 0.001$; ****$P < 0.0001$).

the most recent exposure to parasites expressing VAR2CSA[53]. This stable response is similar to HIV- and cytomegalovirus-specific responses but markedly different from initial SARS-CoV-2 responses, in which IgG is only afucosylated for a few weeks after seroconversion in most patients[34]. This may suggest that the initial SARS-CoV-2-specific antibodies were either secreted by short-lived plasma cells/plasmablasts or that afucosylation in those cells is reprogrammed by particular inflammatory conditions.

**Subunit VAR2CSA vaccination does not induce afucosylated IgG.** When measured at the time of delivery, high levels of IgG recognizing placenta-sequestering IEs are strongly associated with protection from adverse pregnancy outcome[9,23,56]. Many of these antibodies interfere with placental IE sequestration[21,22], and it is therefore generally assumed that neutralizing (adhesion-blocking) antibodies are required for clinical protection against PM[57–59]. On this basis, development of vaccines to prevent PM, based on the so-called minimal-binding-domain (MBD) of VAR2CSA, is currently in progress[60,61]. To examine the levels of Fc fucosylation of VAR2CSA-specific IgG following subunit vaccination, we tested plasma samples from the PAMVAC Phase 1 clinical trial, which involved adult volunteers without previous *P. falciparum* exposure, vaccinated with a recombinant VAR2CSA-MBD construct[60]. In contrast to the results obtained with naturally induced VAR2CSA IgG1, the PAMVAC vaccination induced almost completely fucosylated IgG1; even significantly more fucosylated than total plasma IgG from the same donors (Fig. 4a). The remaining glycosylation traits showed changes of antigen-specific IgG compared to total IgG, similar to naturally infected subjects (Supplementary Fig. 1G–I). This is in line with our recent comparison of naturally acquired and subunit vaccine-induced IgG1 specific for hepatitis B virus[34]. To assess the possibility that the full fucosylation of the vaccine-induced VAR2CSA-specific IgG was due to the vaccinees' lack of previous exposure to *P. falciparum*, genetics, or other environmental parameters, we also tested samples obtained from the parallel trial of the PAM-VAC vaccine in Beninese nulligravidae, who were therefore unexposed to VAR2CSA despite lifelong *P. falciparum* exposure. The results (Fig. 4b and Supplementary Fig. 1J–L) were essentially identical to those obtained with unexposed volunteers. Similar to the Ghanaian cohorts described above, the Beninese cohort had lower Fc fucosylation levels of total plasma IgG compared to previous reports of European cohorts and the unexposed vaccine cohort consisting of Europeans, reaffirming previous reports from rural areas[62]. This is likely due to accumulating afucosylated IgG to both *P. falciparum* membrane antigens and enveloped viruses[34,37].

previous findings regarding fucosylation of IgG1 alloantibodies being stable for >10 years[41,46]. However, unlike the Fc afucosylation of PfEMP1-specific IgG1, which appeared to be exposure-dependent, boosting with alloantigens was found to have no apparent effect on the Fc fucosylation[41,46]. It also suggests that in these cases, afucosylated IgG1 is secreted by long-lived plasma cells, which for VAR2CSA are sustained for up to a decade after

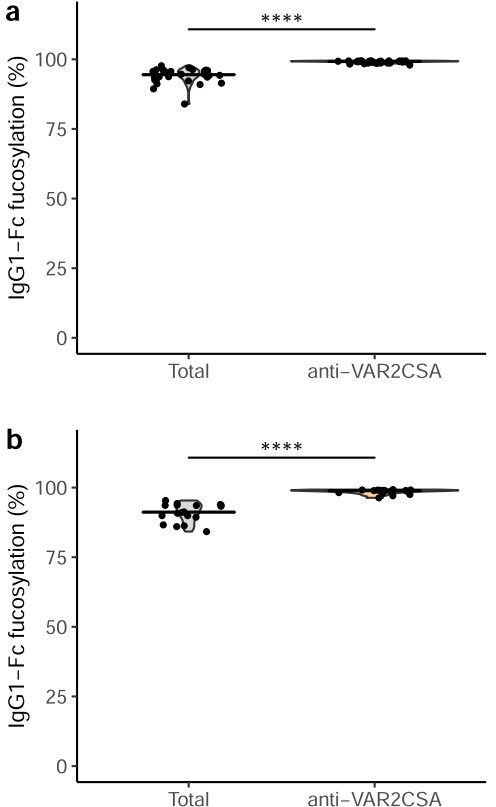

**Fig. 4 VAR2CSA-specific IgG induced by subunit vaccination is not Fc afucosylated.** Fc fucosylation levels of total (gray) and VAR2CSA-specific (orange) plasma IgG1 in German vaccinees (*n* = 32) without (**a**) and in Beninese vaccinees (*n* = 18) with (**b**) natural exposure to *P. falciparum*. Medians and densities are shown. Statistically significant pairwise differences between antigen-specific IgG and total IgG (two-sided Wilcoxon signed-rank test) are indicated (****$P$ < 0.0001).

**Only afucosylated VAR2CSA-specific IgG induces natural killer cell degranulation**. Afucosylation of IgG Fc improves the affinity of IgG for FcγRIII[32,35], increasing NK-cell-mediated ADCC against IgG-opsonized targets[36]. Recently it was reported that IgG from individuals naturally exposed to *P. falciparum* makes IEs susceptible to NK-cell-mediated ADCC, and that PfEMP1-specific IgG is a major contributor to this response[29]. To investigate the functional importance of afucosylation of PfEMP1-specific IgG for ADCC, we assessed the ten Ghanaian plasma samples with the highest and lowest Fc fucosylation of VAR2CSA-specific IgG (Fig. 5a), respectively, for NK-cell degranulation efficiency. The samples had similar VAR2CSA-specific IgG levels ($P$ = 0.80, Mann–Whitney test; Fig. 5b). However, only VAR2CSA-specific IgG from individuals with low VAR2CSA-specific Fc fucosylation caused marked NK-cell degranulation-induced expression of CD107a ($P$ = 0.0003, Mann–Whitney test; Fig. 5c). The only afucosylated VAR2CSA IgG1 not stimulating strong NK-cell degranulation was the sample with the lowest IgG levels. In line with earlier work[35,36], the fucosylation status of these antibodies proved to be a more important predictor of NK-cell-mediated activity than their quantity (Fig. 5b). Apart from fucosylation, only anti-VAR2CSA IgG1-Fc galactosylation correlated with NK-cell degranulation ($r$ = 0.44, $P$ = 0.05). However, this was only borderline statistically significant, reaffirming previous observations of slightly increased FcγRIII affinity of afucosylated IgG with increased Fc galactosylation[35]. To consolidate these results and to directly compare the impact of Fc fucosylation, we next assayed

recombinant fucosylation variants of the VAR2CSA-specific human monoclonal antibody PAM1.4. Whereas both bound VAR2CSA similarly in ELISA (Fig. 5d), only the afucosylated PAM1.4 induced marked NK-cell degranulation (Fig. 5e). Together, these findings underscore the functional significance of Fc afucosylation of PfEMP1-specific IgG, indicating that IgG induced by PfEMP1 subunit vaccination lack potentially important characteristics of the naturally acquired antibody response.

Our study supports the hypothesis that the immune system has evolved a capacity to selectively modulate the glycosylation pattern of the IgG Fc region, thereby fine-tuning the effector response triggered by antibody-opsonized targets[34]. Specifically, it appears that immunogens expressed on host membranes induce afucosylated IgG, thereby increasing its ability to elicit FcγRIII-dependent effector responses such as ADCC. In contrast, immunogens in solution or present on the surface of pathogens seem to mainly induce fucosylated IgG, thus steering the effector response against IgG-opsonized targets towards other FcγR-dependent effector functions. The plasticity in human immune responses to modulate IgG effector functions by altered fucosylation endows the immune system with a so far largely unappreciated level of adaptability. While it is congruent with the current understanding of how the immune system works, the functional importance of afucosylated IgG in malaria remains to be demonstrated. In the meantime, it should be emphasized that the decrease in Fc fucosylation reported here exceeds any that has previously been reported for pathogen-derived antigens. Indeed, it also surpasses the clinically significant afucosylation of the IgG response to alloantigens[39,40,46] found in pregnancy where mothers are naturally exposed to paternally inherited antigens on fetal cells such as blood cells. These antigens are, in essence, foreign to the maternal immune system and exposed on the cell surface in a way akin to truly foreign pathogen-derived antigens by intracellular pathogens expressing proteins on the host surface such as enveloped viruses[34] and *P. falciparum*. This implies that the immunopathogenic IgG raised in alloimmune-mediated diseases in pregnancies is an unfortunate corollary of an evolutionary conserved and advantageous immune mechanism against intracellular pathogens, raising potent afucosylated IgG needed to eliminate these. Several immunomodulatory molecules are likely to regulate this mechanism, but due to the apparent necessity of self-membrane association of the antigen, we have previously proposed a model with a yet unknown membrane-bound receptor on B cells recognizing self[34]. Signaling from this receptor is likely to result in transcriptional changes, either decreasing the expression of fucosyltransferases or increasing the expression of molecular chaperones limiting the access of fucosyltransferases to the N297 glycan. Nevertheless, the mechanisms of this immunomodulation are beyond of the scope of this study. Finally, the data suggest that to induce afucosylated IgG responses with increased ADCC—and potentially protective capacity—alternative vaccination strategies are required, mimicking the expression of antigens on host cells.

## Methods

**Human subjects**. We used biological samples collected as part of the following studies: (i) A longitudinal study of malaria in pregnancy, conducted in Dodowa, located in a coastal savannah area with stable, seasonal *P. falciparum* transmission, approximately 40 km north of Accra, Ghana[48]. (ii) A cross-sectional study of immune responses to VAR2CSA in healthy non-pregnant women[53], conducted in Assin Foso, in a rainforest area with high and perennial *P. falciparum* transmission, located approximately 80 km north of Cape Coast, Ghana[63]. (iii) A phase 1 clinical trial of the VAR2CSA-based PAMVAC vaccine, conducted in non-immune German volunteers and in adult, nulligravid *P. falciparum*-exposed Beninese women volunteers[60]. Healthy blood donor samples from Sanquin, Amsterdam, The Netherlands, were used as negative control donors.

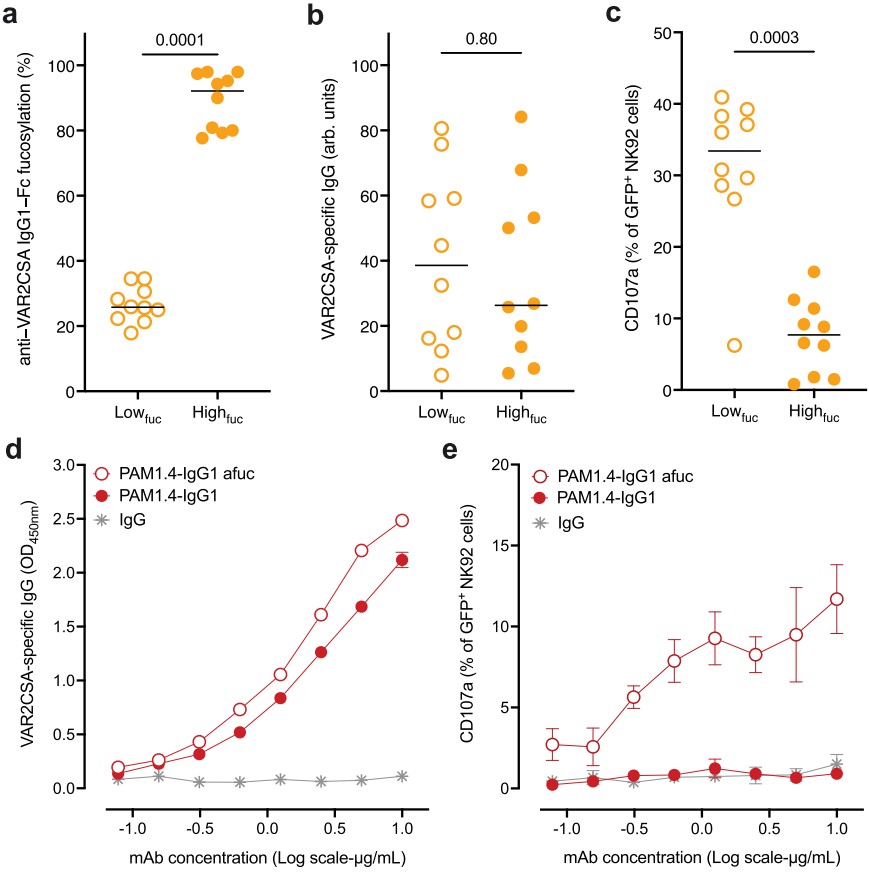

**Fig. 5 Afucosylated PfEMP1-specific IgG induces NK-cell-mediated ADCC. a** VAR2CSA-specific IgG fucosylation levels of samples with low (open symbols) and high fucose (filled symbols) levels. **b** Comparison of VAR2CSA-specific IgG levels and **c** CD107a expression between highly fucosylated (filled symbols) and afucosylated anti-VAR2CSA IgG1 (open symbols) samples, respectively. Mean from three independent experiments and P values from two-sided Mann–Whitney tests are shown. **d** VAR2CSA-specific, human monoclonal antibody PAM1.4 as either fucosylated or afucosylated IgG1 was titrated in the same assay and VAR2CSA-binding or **e** degranulation activity (CD107a expression) on NK92-CD16a cells was measured. IgG: non-immune human IgG. Data represent mean values ± SD from three independent experiments.

The Ghanaian donors all had serologic evidence of exposure to *P. falciparum*, with seropositivity rates above 90% in the non-pregnant cohort[53] and above 70% in the pregnant cohort[48].

A more detailed demographic description of the analyzed cohorts can be found in the Supplementary Information (Supplementary Table 3).

***P. falciparum* recombinant antigens**. The full-length ectodomains of the VAR2CSA-type PfEMP1 antigen IT4VAR04 (VAR2CSA) and of the Group A PfEMP1 antigen HB3VAR6 (VAR6) were expressed in baculovirus-infected insect cells and purified as described previously[58,64]. The amino-terminal, non-repetitive R0 region of glutamate-rich protein (GLURP) was expressed in *Escherichia coli* and purified as described elsewhere[65].

**Purification of IgG from plasma samples**. Total IgG from individual donors was purified from ~1 µL plasma using the AssayMAP Bravo platform (Agilent Technologies, Santa Clara, USA) with Protein G-coupled cartridges as described elsewhere[34].

*P. falciparum* antigen-specific IgG was purified from individual donors by incubation (1 h, room temperature) of individual plasma samples (diluted 1:10 in phosphate-buffered saline (PBS) supplemented with TWEEN 20 (0.05 %; PBS-T)) in 96-well Maxisorp plates (Nunc, Roskilde, Denmark) coated overnight (4 °C; PBS) with VAR2CSA (2 µg/mL), VAR6 (2 µg/mL), or GLURP (1 µg/mL). Following the incubation, the plates were washed 3× with PBS-T, 2× with PBS, and 2× with ammonium bicarbonate (50 mM). Antigen-specific IgG were finally eluted by formic acid (100 mM; 5 min shaking).

**Mass spectrometric IgG Fc glycosylation analysis**. Eluates of purified IgG were collected in low-binding PCR plates (Eppendorf, Hamburg, Germany) and dried by vacuum centrifugation (50 °C). The dried samples were dissolved in a reduction and alkylation buffer containing sodium deoxycholate (0.4%), tris(2-carboxyethyl) phosphine (10 mM), 2-chloroacetamide (40 mM), and TRIS (pH 8.5; 100 mM), or ammonium bicarbonate (50 mM). After boiling the samples (10 min; 95 °C), trypsin (5 µg/mL) in ammonium bicarbonate (50 mM) was added. The digestion

was terminated after overnight incubation (37 °C) by acidifying to a final concentration of 2% formic acid. Prior to mass spectrometry injection, sodium deoxycholate precipitates, in samples where this was added, were removed by centrifugation (3000 × g; 30 min), and filtering through 0.65 µm low protein binding filter plates (Millipore, Burlington, USA).

Analysis of IgG Fc glycosylation was performed with nanoLC reverse phase-electrospray mass spectrometry on an Impact HD quadrupole-time-of-flight mass spectrometer (Bruker Daltonics, Bremen, Germany), and data were processed with Skyline software (version 4.2.0.19107) as described elsewhere[34]. Distinct samples were measured once. Samples were considered seropositive if the intensity of antigen-specific IgG1 glycopeptides was at least higher than the mean plus 10× the standard deviation of Dutch seronegative control samples. The level of fucosylation and bisection were calculated as the sum of the relative intensities of glycoforms containing the respective glycotraits. Galactosylation and sialylation levels were calculated as antenna occupancy. The relative intensities of the glycoforms were summed with mono-galactosylated/sialylated species only contributing with 50% of their relative intensity. Details on analyzed glycopeptides can be found in the Supplementary Information (Supplementary Table 4)

**Human monoclonal VAR2CSA-specific IgG**. The human monoclonal IgG1 antibody, PAM1.4, derived from an EBV-immortalized memory B-cell clone from a Ghanaian woman with natural exposure to PM[66], recognizes a conformational epitope in several VAR2CSA-type PfEMP1 proteins, including IT4VAR04. In the present study, we used a non-modified recombinant version of PAM1.4 produced in HEK293F cells with high Fc fucosylation and a glyco-engineered variant with low Fc fucosylation[26,67].

**Quantification of VAR2CSA-specific IgG**. Levels of VAR2CSA-specific IgG were assessed by ELISA as previously described[68]. In brief, 96-well flat-bottom microtiter plates (Nunc MaxiSorp, Thermo Fisher Scientific) were coated overnight at 4 °C with full-length VAR2CSA (100 ng/well in PBS). Monoclonal antibody (0.08–10 µg/mL) or plasma samples (1:400) were added in duplicate, followed by washing and

horseradish peroxidase-conjugated rabbit anti-human IgG (1:3,000; Dako). Bound antibodies were detected by adding TMB PLUS2 (Eco-Tek), and the reaction stopped by the addition of 0.2 M $H_2SO_4$. The optical density (OD) was read at 450 nm and the specific antibody levels were calculated in arbitrary units (arb. units), using the equation $100 \times [(OD_{SAMPLE} - OD_{BLANK})/(OD_{POS.CTRL} - OD_{BLANK})]$.

**Antibody-dependent cellular cytotoxicity (ADCC) assay.** Degranulation-induced CD107a expression in response to IgG bound to plastic-immobilized antigen is a convenient marker of NK-cell ADCC[69]. Here, we coated 96-well flat-bottom microtiter plates (Nunc MaxiSorp; Thermo Fisher Scientific) overnight at 4 °C with full-length VAR2CSA (100 ng/well in PBS[68]). Following 1 h blocking with PBS containing 1% Ig-free bovine serum albumin (BSA) (1% PBS-BSA), plasma samples (1:20) or PAM1.4 variants (0.08–10 µg/mL) were added for 1 h at 37 °C. After washing, $1.6 \times 10^5$ NK92 cells stably expressing CD16a and GFP[70] were added to each well. In addition, anti-human CD107a-PE (1:40; H4A3 clone; BD Biosciences), 10 µg/mL brefeldin A (Sigma-Aldrich), and 2 µM monensin (Sigma-Aldrich) were added, and the cells incubated for 4 h at 37 °C. Cells were then centrifuged and stained with near-IR fixable Live/Dead dye (Invitrogen), followed by data acquisition on a FACS LSRII flow cytometer (BD Biosciences), and analysis with FlowLogic software (version8.3; Inivai Technologies, Australia). Wells with antigen and NK cells, but without antibody were included in all experiments to control for unspecific activation. Plasma samples from four Danish non-pregnant women without malaria exposure and purified human IgG (Sigma-Aldrich) were included as negative controls. Example of the gating strategy is provided as Supplementary Information (Supplementary Fig. 6)

**Statistical tests.** Statistical analyses were performed using R: A Language and Environment for Statistical Computing (Version 3.5.2). Performed tests are mentioned in the text. All performed tests were two-sided and tests for normal distribution of data were performed where applicable.

**Ethics statement.** Collection of biological samples for this study was approved by the Institutional Review Board of Noguchi Memorial Institute for Medical Research, University of Ghana (study 038/10-11) by the Regional Research Ethics Committees, Capital Region of Denmark (protocol H-4-2013-083), by the Academic Medical Center Institutional Medical Ethics Committee of the University of Amsterdam, by the Ethics Committee of the Medical Faculty and the University Clinics of the University of Tubingen, and by the German Regulatory authorities. The study was conducted in adherence to the International Council for Technical Requirements for Human Use guidelines and the principles of the Declaration of Helsinki. Written informed consent was obtained from all participants before enrollment.

**Reporting summary.** Further information on research design is available in the Nature Research Reporting Summary linked to this article.

## Data availability

The liquid chromatography mass spectrometry data generated in this study have been deposited in the MassIVE database under accession code MSV000088060. Further data that support the findings of this study are available from the corresponding author upon reasonable request. Data regarding the original vaccination study are available from ClinicalTrial.gov ID: NCT02647489. Source data are provided with this paper.

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

## Acknowledgements

We are grateful to all the individuals donating blood samples for this study, and to the scientists and health workers participating in the studies for which they were originally collected. We thank Michael Theisen (University of Copenhagen and Statens Seruminstitut) for GLURP antigen and GLURP-reactive IgG preparation, and Bruce Walcheck and Geoff Hart (University of Minnesota) for the NK92-CD16a cell line. We also acknowledge Erik de Graaf (Sanquin Research) for optimization of the analysis pipelines used in this study, done in relation to previous projects. The study was funded by the Landsteiner Foundation for Blood Transfusion Research grant number 1721 and the Danish International Development Agency (Danida), 12-081RH and 17-02-KU. The PAMVAC study (ClinicalTrials.gov ID NCT02647489) was sponsored by the Universitätsklinikum Tübingen and funded by the European Union Seventh Framework Programme (FP7-HEALTH-2012-INNOVATION; under grant agreement 304815), the Danish Advanced Technology Foundation (under grant number 005-2011-1), and a Medium Scale Collaborative Project supported by the German Federal Ministry of Education and Research (Bundesministerium für Bildung und Forschung) through EVI, KfW, and Irish Aid. The funders had no role in study design, data collection and analysis, decision to publish, or preparation of the manuscript.

## Author contributions

M.D.L., C.E.v.dS., L.H., and G.V. conceptualized the study. M.F.O., L.H., and G.V. acquired funding. M.D.L., M.L.-P., J.N., M.W., L.H., and G.V. performed the investigations presented in this study. E.K.D., P.A., B.M., A.S., M.A.N., M.F.O., N.T.N., and A.M. supplied study material. M.D.L., L.H., and G.V. wrote the original draft which was reviewed and edited by all authors.

## Competing interests

The authors declare no competing interests.
