## [Peer Review File · Nature Communications]

Reviewer comments, initial round of review: - -

Reviewer #1 (Remarks to the Author):

Larsen and Lopez-Perez and colleagues show differences in fucosylation of anti-var2CSA IgG produced in response to natural infection versus VAR2CSA-type subunit vaccine, which lead to different ability to induce NK-cell ADCC. The data is relevant for the broad malaria vaccine and immunology community's it highlights the importance of membrane antigens being presented in their natural structure.

I propose a few alterations and additions to the manuscript that would in my view increase the comprehension and reach of the data.

major points:

the authors show in Fig 2B fucosylation of var2CSAIgG decreases with increasing parity and I recommend showing also how the data would look when plotting the level of VAR2CSA-specific IgG1 fucosylation against age instead of parity. if the correlation is absent or less pronounced it would further support the authors conclusion that it is increased exposure decreasing fucosylation.

I also think the readers would gain from knowing how levels of VAR2CSA and VAR6 IgG1 fucosylation relates to titers of VAR2CSA and VAR6 IgG1 in the Ghanaian women.

The range of VAR2CSA IgG1 fucosylation is very broad even in women with low number of pregnancies, and no mention is made to the protection level associating with each woman, and if indeed higher fucosylation of var2CSA IgG leaves women at higher risk of PM complication, and what is the relative importance of NK dependent ADCC in protection from negative effects of *P. falciparum* infection during pregnancy. I suggest discussing.

minor points:

in my view the paper would gain from having the result and discussion in separated sections, but I leave it to the editor to make a better judgement of this suggestion.

I suggest adding a short sentence to introduce fucose and fucosylation of Ig around lines 105-106 could be of help touring readers less acquainted to the topic into context

in line 172 suggesting is probably better than indicating

Fig 2B-E seem to have data from women that are not present in all graphs (clearly for women with 5 and 9 previous parities) I suggest revising, in case that this is a mistake, or including only data from women having all analyses done.

line 205 it is not clear to me to what "those AB" refer to, tho HIV and cytomegalovirus or SARS-CoV-2, i recommend clarifying.

Sup fig. 1G-L are not properly called in the text as the sentences where they are included refer only to fucosylation and mention not Galactosylation, Sialylation or bisecting. I recommend revising.

Reviewer #2 (Remarks to the Author):

The manuscript of Larsen et al describes a markedly and significant increase in afucosylated IgG in response to infection with the malaria causing parasite *P.falciparum*, in contrast to natural IgG which is almost fully fucosylated. In fact the increase in afucosylated IgG following *P.falciparum* infection is substantially more than what is observed in other infections such as HIV infection. Interestingly, it is also reported that this increase in afucosylated IgG is not observed following subunit (VAR2CSA) vaccination, a finding that has potential implications for future vaccination strategies. This decrease in mainly fucosylated IgG1 is stable and persistent, according to the study. Furthermore, it is also reported that other glycoforms of IgG are increased, such as galactosylation and sialylation. Taken together, these results suggest that following *P.falciparum* infection the immune system produces IgG which recognise membrane bound antigens of infected erythrocytes and have higher affinity for FcγRIIIa in particular due to the specific glycoforms (mainly lack of core fucose) on Asn297 in the Fc region of IgG1, following opsonisation and immune complex formation. This in turn leads to an increase in NK cell mediated ADCC and ADCP. NK degranulation induced only with afucosylated subunit (VAR2CSA) specific IgG is shown, in contrast to fucosylated IgG and the MAb Pam1.4. These results are important and novel in terms of *P.falciparum* infection and malaria and subunit vaccination. While it is not novel that afucosylated IgG is increased in certain disease conditions and infections, as stated by the authors during viral infection and pregnancy and it is well established that this IgG1 glycoform preferentially activates NK cells to induce ADCC and ADCP through increased affinity for FcγRIIIa, it is novel that this occurs following *P.falciparum* infection and is possibly a factor in the antibody immune response to malaria.

The hypothesis proposed by the authors is that increases in afucosylated IgG only occurs in response to foreign antigens such as viruses (membrane bound antigens of enveloped viruses) and alloantigens. Results are presented in favour of this hypothesis such as increases in afucosylation in response to the pfEMP1 antigens Var6 and Var2CSA. Also increases in afucosylated IgG were not observed following vaccination with the VAR2CSA subunit vaccine and only afucosylated IgG specific for the antigen VAR2CSA induces natural killer degranulation through increased affinity for FcγRIIIa. Also these results support the hypothesis that afucosylation is attained following repeat exposure and this afucosylation is retained and is stable and persistent, as stated by the authors.

Of more broad interest the authors propose a hypothesis where the immune system has preferentially evolved to produce and modify IgG glycosylation in the Fc region through increased afucosylation in particular to improve antibody binding to FcγRs of immune cells such as NK cells and fine tune the immune response. While the results presented here and elsewhere point to such a mechanism the data is very much correlatory, where increases in afucosylation correlate with certain types of infection. A mechanism for this fine tuning by the immune system is not proposed or discussed. I think discussion of this process or a proposed mechanism might help, where the immune system could achieve this fine tuning through selective expression of glycan processing enzymes or some other mechanism.

The reviewer has no issues with the data, the samples or the analysis. The authors are experts in antibody glycosylation and glycan analytics. There are no issues with the statistical analysis or the conclusions which support the data, just the lack of evidence or mechanism for the fine tuning hypothesis.

I believe the results will be of interest to the wider field. It is not new that increased afucosylated variants of IgG exist in certain conditions, as discussed by the authors in viral infection for example, however is not shown previously in *P.falciparum* or parasite infection and malaria, to my knowledge. It is also of wider interest that this could be an evolutionary mechanism to produce increased IgG with lacks core fucose, even if there is data to support this claim but a lack of explanation as to how it is achieved by the immune system. Supporting evidence for a mechanism is likely outside the scope of this article/study, however.

To this reviewer, the results/data are convincing, the samples/size are scientifically and statistically sound, the analysis sound and the conclusions of the experiments convincing.

I believe the experiments and results could be repeated given access to the same samples with the same level of detail provided in the study.

Reviewer #3 (Remarks to the Author):

Larsen/Lopez-Perez and colleagues present an interesting and important study of the acquisition of afucosylation of PfEMP1 antibodies and the potential impact this may have on function. Overall this is a relevant and significant study in an overlooked field. I have the following suggestions to improve the manuscript, but consider the data of relevance and importance, and suitable for publication.

Major comment: The lack of a cohort study to show that fucosylation levels impacts protection is of note. While this limitation is already mentioned in discussion, the Ofori et al 2009 cohort has previously been used to look at malaria pregnancy outcomes, so it is unclear why no protection analysis with outcomes has been explored here.

Minor comments:

Results:

Figure 2, Line 165 – is the difference in Fc fucosylation between VAR2CSA and VAR6 statistically significant. I was unclear from figure legend if all pairwise comparisons were made, or just between total and each antigen specific Ab.

Figure 3 – 4 women with no previous pregnancies – would expect very low levels (non?) VAR2CSA Ab in these women. What explains these data? Are these unrecorded previous pregnancies? Or Antibodies that are induced by some low level of cross reactive PfEMP1? What is the OD of the VAR2CSA? Does it suggest a specific response or cross reaction? Would it make more sense to remove these nulligravidae women from the analysis?

Lines 186 – “afucosylation...more pronounced ...”. It is hard for the reader to assess this statement as the data isn't presented side by side. Please add a supplementary Figure, and statistical analysis comparing the two groups.

Further – Why would the VAR2CSA afucosylation be impacted by transmission? My understanding is that VAR2CSA infection occurs predominately (exclusively?) during pregnancy? So that if afucosylation would be driven by exposure, then the levels of afucosylation will be parity dependent, but not dependent on the overall transmission intensity. Can these factors be explored in a multivariate linear regression of the two cohorts and pregnancy/afucosylation levels? If the afucosylation is dependent on overall transmission AND parity, what mechanism may underpin this?

Figure 5

This section is a bit clunky, and would suggest an edit. It is quite confusing having CD107a/Ab level and Fc fucosylation switched on the axis A/B – would suggest modify for ease of reading. My interpretation of the data is that – 1) IgG levels are the same between low/high fucosylation. 2) Level of fucosylation strongly associated with degranulation (but total IgG level is not). If this is the main point authors trying to make, I suggest an edit of text/figure.

Line 244 – levels of VAR2CSA IgG between groups – can you include a statistical analysis? And also is this total IgG? A comparison of IgG1 would be more informative.

Line 252 – Is there any evidence that the detection of PAM1.4 in ELISA can be impacted by secondary/detection antibodies be impacted by afucosylation levels?

Do any of the other antibody characteristics (galactosylation/sialylation/bisection GlcNAc – as in

Supplementary Figure) also associated/correlate with NK cell degranulation? Are these characteristics all associated with afucosylation levels?

Discussion

There is a lot of discussion in the results sections that could be moved to discussion to make a longer discussion/conclusion

Methods queries

Is the purity of the antigen specific IgG used for mass spec known?

REVIEWER COMMENTS

Reviewer #1 (Remarks to the Author):

Larsen and Lopez-Perez and colleagues show differences in fucosylation of anti-var2CSA IgG produced in response to natural infection versus VAR2CSA-type subunit vaccine, which lead to different ability to induce NK-cell ADCC. The data is relevant for the broad malaria vaccine and immunology community's it highlights the importance of membrane antigens being presented in their natural structure.

I propose a few alterations and additions to the manuscript that would in my view increase the comprehension and reach of the data.

major points:

the authors show in Fig 2B fucosylation of var2CSA IgG decreases with increasing parity and I recommend showing also how the data would look when plotting the level of VAR2CSA-specific IgG1 fucosylation against age instead of parity. if the correlation is absent or less pronounced it would further support the authors conclusion that it is increased exposure decreasing fucosylation. *Age and parity are strongly correlated, and unsurprisingly, fucosylation of VAR2CSA-specific IgG1 showed a relation to age (new Supplementary Fig. 2A), which was similar to parity relation (Fig. 2B). As expected, there was no significant relationship between fucosylation of the non-pregnancy associated VAR6-specific IgG and parity (Fig. 2C), and its relation to age (Supplementary Fig. 2B) was much weaker than for VAR2CSA. Fucosylation of GLURP-specific IgG did not correlate with neither parity (Fig. 2D) nor age (Supplementary Fig. 2C). Together, these data very strongly indicate that age per se is of minor importance, contributing only a very limited decrease, evidenced by the analysis of fucosylation of total IgG (Fig. 2E and Supplementary Fig. 2D). This is discussed in lines 186-8 and lines 192-5 of the revised manuscript in "Show all markup" view:*

"This is further supported by the fact that VAR6-specific IgG-Fc fucosylation was correlated with age, as was total IgG-Fc fucosylation, similar to previous reports (Supplementary Fig. 2)."

"VAR2CSA-specific IgG Fc fucosylation was also negatively correlated with age (Supplementary Fig. 2A), most likely reflecting the strong correlation between parity and age. In contrast, no correlation was found between age and GLURP-specific IgG Fc fucosylation (Supplementary Fig. 2)."

I also think the readers would gain from knowing how levels of VAR2CSA and VAR6 IgG1 fucosylation relates to titers of VAR2CSA and VAR6 IgG1 in the Ghanaian women.

We have added a new supplementary figure (Supplementary Fig. 3) showing that there were no correlations between titers and Fc fucosylation level. The figure is referred to in line 200 of the revised manuscript in "Show all markup" view:

"No correlations between IgG levels and Fc fucosylation levels were observed (Supplementary Fig. 3)."

The range of VAR2CSA IgG1 fucosylation is very broad even in women with low number of pregnancies, and no mention is made to the protection level associating with each woman, and if indeed higher fucosylation of var2CSA IgG leaves women at higher risk of PM complication, and what is the relative importance of NK dependent ADCC in protection from negative effects of P.

falciparum infection during pregnancy. I suggest discussing.

The wide range of fucosylation of VAR2CSA-specific IgG1 (Fig. 2B) is highly likely to reflect the stochastic nature of placental malaria, i.e., the fact that only some women will suffer PM in any given pregnancy. We therefore predict a decreasing range with increasing parity, but we did not have access to a sufficient number of high-parity women to assess this hypothesis directly.

Prompted by this reviewer comment and the comments by reviewer 3 (see below), we have included an analysis of the putative protective impact of the observed selective afucosylation of VAR2CSA-specific IgG in the revised manuscript (Supplementary Tables 1 and 2). The results, which must be considered preliminary due to the relatively small number of available data points, approach conventional statistical significance. This is encouraging given the above-mentioned stochastic nature of PM, and suggest that a more definitive analysis in a future, bigger study is warranted. We have highlighted this point in the revised manuscript. In the meantime, the data in Supplementary Tables 1 and 2, and the data on the impact of afucosylation on NK cell-mediated ADCC (Fig. 5) strongly suggest the existence of a clinical impact of afucosylation. If indeed it exists, it would have important consequences for vaccination strategies, and we also discuss that point in the revised manuscript.

minor points:

In my view the paper would gain from having the result and discussion in separated sections, but I leave it to the editor to make a better judgement of this suggestion.

We fully acknowledge that this of course is a matter of preference. However, we prefer to maintain the current format, which has been used effectively in other Nat Commun papers (see Wang et al. Nat Commun 12, 2956, 2021 for a recent example). However we are of course open to splitting this section if the editor prefers.

I suggest adding a short sentence to introduce fucose and fucosylation of Ig around lines 105-106 could be of help touring readers less acquainted to the topic into context

We have added a sentence introducing how different saccharides are added to the core structure of the Fc-glycan (lines 111-6 in "Show all markup" view), along with a reference to figure 1C that depicts the structure of the Fc-glycan:

"Different monosaccharides, such as galactose, fucose, and sialic acids, are added to the bi-antennary core structure of the Fc-glycan (Fig. 1C). The level of fucosylation is of particular significance, since afucosylated IgG has up to 20-fold increased affinity for FcγR11a. The reason is believed to be a steric clash between a unique N-linked glycan at position N162, not found in FcγR1a nor FcγR11a/b, and the bi-antennary glycan in the IgG-Fc imposed by the core-fucose"

in line 172 suggesting is probably better than indicating

This has now been changed.

Fig 2B-E seem to have data from women that are not present in all graphs (clearly for women with 5 and 9 previous parities) I suggest revising, in case that this is a mistake, or including only data from women having all analyses done.

As the reviewer rightly points out, a few data points are indeed missing from Fig. 2, due to inconclusive data from the glycosylation analysis. However, this has no impact on the results presented, as we do not attempt analysis at the individual level. It is therefore not required – or indeed justified – to remove all data points for women where the data point for a single antigen-specificity is missing. To alleviate the concern regarding the number of missing data points, we have added this information in Fig. 2.

line 205 it is not clear to me to what “those AB” refer to, tho HIV and cytomegalovirus or SARS-CoV-2, i recommend clarifying.

We agree that this was sentence was unclear. We have now clarified the sentence by defining “those AB” as SARS-CoV-2-specific antibodies (lines 239-41 in “show all markup” view):

“This may suggest that those the initial SARS-CoV-2-specific antibodies were either secreted by short-lived plasma cells/plasmablasts, or that afucosylation in those cells is reprogrammed by particular inflammatory conditions”

Sup fig. 1G-L are not properly called in the text as the sentences where they are included refer only to fucosylation and mention not Galactosylation, Sialylation or bisecting. I recommend revising.

We agree that these panels were not properly mentioned in the next. A sentence (lines 257-9 in “Show all markup” view) has now been added mentioning panel sup fig. 1G-I, which also clarifies sup fig. 1J-L (line 265)

“The remaining glycosylation traits showed changes of antigen-specific IgG compared to total IgG, similar to naturally infected subjects (Supplementary Fig. 1G-I)”

Reviewer #2 (Remarks to the Author):

The manuscript of Larsen et al describes a markedly and significant increase in afucosylated IgG in response to infection with the malaria causing parasite *P.falciparum*, in contrast to natural IgG which is almost fully fucosylated. In fact the increase in afucosylated IgG following *P.falciparum* infection is substantially more than what is observed in other infections such as HIV infection. Interestingly, it is also reported that this increase in afucosylated IgG is not observed following subunit (VAR2CSA) vaccination, a finding that has potential implications for future vaccination strategies. This decrease in mainly fucosylated IgG1 is stable and persistent, according to the study. Furthermore, it is also reported that other glycoforms of IgG are increased, such as galactosylation and sialylation. Taken together, these results suggest that following *P.falciparum* infection the immune system produces IgG which recognise membrane bound antigens of infected erythrocytes and have

higher affinity for FcγRIIIa in particular due to the specific glycoforms (mainly lack of core fucose) on Asn297 in the Fc region of IgG1, following opsonisation and immune complex formation. This in turn leads to an increase in NK cell mediated ADCC and ADCP. NK degranulation induced only with afucosylated subunit (VAR2CSA) specific IgG is shown, in contrast to fucosylated IgG and the MAb Pam1.4. These results are important and novel in terms of *P.falciparum* infection and malaria and subunit vaccination. While it is not novel that afucosylated IgG is increased in certain disease conditions and infections, as stated by the authors during viral infection and pregnancy and it is well established that this IgG1 glycoform preferentially activates NK cells to induce ADCC and ADCP through increased affinity for FcγRIIIa, it is novel that this occurs following *P.falciparum* infection and is possibly a factor in the antibody immune response to malaria.

The hypothesis proposed by the authors is that increases in afucosylated IgG only occurs in response to foreign antigens such as viruses (membrane bound antigens of enveloped viruses) and alloantigens. Results are presented in favour of this hypothesis such as increases in afucosylation in response to the pfEMP1 antigens Var6 and Var2CSA. Also increases in afucosylated IgG were not observed following vaccination with the VAR2CSA subunit vaccine and only afucosylated IgG specific for the antigen VAR2CSA induces natural killer degranulation through increased affinity for FcγRIIIa. Also these results support the hypothesis that afucosylation is attained following repeat exposure and this afucosylation is retained and is stable and persistent, as stated by the authors.

Of more broad interest the authors propose a hypothesis where the immune system has preferentially evolved to produce and modify IgG glycosylation in the Fc region through increased afucosylation in particular to improve antibody binding to FcγRs of immune cells such as NK cells and fine tune the immune response. While the results presented here and elsewhere point to such a mechanism the data is very much correlatory, where increases in afucosylation correlate with certain types of infection. A mechanism for this fine tuning by the immune system is not proposed or discussed. I think discussion of this process or a proposed mechanism might help, where the immune system could achieve this fine tuning through selective expression of glycan processing enzymes or some other mechanism.

We agree we were perhaps too brief on discussing the proposed mechanism. The reason for this is because we recently included and extensively discussed this proposed mechanism in ref 34. We have now included a brief discussion in the conclusion section (lines 325-31 in "show all markup" view) briefly summing up our previously described hypothesis and stating which changes we hypothesize to find in B cells producing afucosylated IgG.

“Several immunomodulatory molecules are likely to regulate this mechanism, but due to the apparent necessity of self-membrane association of the antigen, we have previously proposed a model with a yet unknown membrane-bound receptor on B cells recognizing self³⁴. Signaling from this receptor is likely to result in transcriptional changes, either decreasing the expression of fucosyltransferases or increasing the expression of molecular chaperones limiting the access of fucosyltransferases to the N297 glycan. Nevertheless, the mechanisms of this immunomodulation are beyond of the scope of this study.”

The reviewer has no issues with the data, the samples or the analysis. The authors are experts in antibody glycosylation and glycan analytics. There are no issues with the statistical analysis or the conclusions which support the data, just the lack of evidence or mechanism for the fine tuning hypothesis.

We are happy to hear that and thank the reviewer for these kind remarks.

I believe the results will be of interest to the wider field. It is not new that increased afucosylated variants of IgG exist in certain conditions, as discussed by the authors in viral infection for example, however is not shown previously in P.falciparum or parasite infection and malaria, to my knowledge. It is also of wider interest that this could be an evolutionary mechanism to produce increased IgG with lacks core fucose, even if there is data to support this claim but a lack of explanation as to how it is achieved by the immune system. Supporting evidence for a mechanism is likely outside the scope of this article/study, however.

We agree that the mechanism behind this phenomena is intriguing. Our overall hypothesis has been described in a previous publication of ours (Larsen et. al; Science (2021);371(6532):eabc8378). We are certain that these results and theoretical framework will stimulate the scientific community to further unravel the underlying molecular mechanism behind afucosylated IgG responses both in malaria and in general, which we agree is beyond the scope of this study.

To this reviewer, the results/data are convincing, the samples/size are scientifically and statistically sound, the analysis sound and the conclusions of the experiments convincing.

I believe the experiments and results could be repeated given access to the same samples with the same level of detail provided in the study.

We are glad to hear that and we thank the reviewer for these kind remarks.

Reviewer #3 (Remarks to the Author):

Larsen/Lopez-Perez and colleagues present an interesting and important study of the acquisition of afucosylation of PfEMP1 antibodies and the potential impact this may have on function. Overall this is a relevant and significant study in an overlooked field. I have the following suggestions to improve the manuscript, but consider the data of relevance and importance, and suitable for publication.

Major comment: The lack of a cohort study to show that fucosylation levels impacts protection is of note. While this limitation is already mentioned in discussion, the Ofori et al 2009 cohort has previously been used to look at malaria pregnancy outcomes, so it is unclear why no protection analysis with outcomes has been explored here.

The reviewer points out two aspects of importance. The Ofori pregnancy cohort (described in Ofori et al. Ghana Med J 43, 13, 2009) was not set up to tackle the current research question. Furthermore, less than half of the samples originally collected for that study remained available for the current study, which is therefore underpowered for analysis of clinical impact. Despite these reservations, we have now included the requested analysis in supplemental tables 1-2. While the results do not reach conventional statistical significance, the level of fucosylation is borderline significant, warranting future, more definitive studies to directly answer this question, not only in pregnancy, but also for protective immunity involving non-pregnancy PfEMP1 proteins. No clinical data that could be used for this type of analysis were collected as part of the non-pregnancy cohort that is also used in the current report (Ampomah et al. Infect Immun 82, 1860, 2014).

Minor comments:

Results:

Figure 2, Line 165 – is the difference in Fc fucosylation between VAR2CSA and VAR6 statistically significant. I was unclear from figure legend if all pairwise comparisons were made, or just between total and each antigen specific Ab.

Pairwise comparisons were only done between total and antigen-specific IgG. This has now been clarified in the figure legend of figure 2 (line 1046)

Figure 3 – 4 women with no previous pregnancies – would expect very low levels (non?) VAR2CSA Ab in these women. What explains these data? Are these unrecorded previous pregnancies? Or Antibodies that are induced by some low level of cross reactive PfEMP1? What is the OD of the VAR2CSA? Does it suggest a specific response or cross reaction? Would it make more sense to remove these nulligravidae women from the analysis?

*As the reviewer correctly points out, high IgG reactivity to VAR2CSA is not expected among children, men, or women, who have never been pregnant. Nevertheless, levels among such individuals with natural *P. falciparum* exposure are usually somewhat higher than levels among completely unexposed control donors from outside Africa. The most likely explanation is the higher IgG synthesis in many African settings, which means that negative cutoffs are best established based on values obtained with samples from *P. falciparum*-exposed adult males rather than with samples from non-African controls (which unfortunately is the norm).*

However, Fig. 3B does not show levels of VAR2CSA-specific IgG, but rather the degree of fucosylation of VAR2CSA-specific IgG. That can be high (indeed, is expected to be high under our hypothesis) even when levels are quite low (as would certainly be expected). Thus, it could be argued that the most

surprising data points in Fig. 3B are the two nulligravidae with marked afucosylation of VAR2CSA-specific IgG. While we have no firm evidence, the most likely explanation is earlier pregnancies that were not reported to the investigating team for one reason or another.

On this basis, we do not believe that removing the four data points from women with no acknowledged previous pregnancies is justified. In any case, removal would have no discernable impact on the analysis or the conclusions based on it.

We have added a sentence (lines 221-3 in “show all markup” view) to clarify this matter:

“In this cohort, we were also able to purify low levels of VAR2CSA-specific IgG from nulligravidae, plausibly caused by unrecorded/undiscovered pregnancies.”

Lines 186 – “afucosylation...more pronounced ...”. It is hard for the reader to assess this statement as the data isn’t presented side by side. Please add a supplementary Figure, and statistical analysis comparing the two groups.

A supplementary figure has now been added for visual clarity (Supplementary Fig. 3) (line 219 in “Show all markup” view)

Further – Why would the VAR2CSA afucosylation be impacted by transmission? My understanding is that VAR2CSA infection occurs predominately (exclusively?) during pregnancy? So that if afucosylation would be driven by exposure, then the levels of afucosylation will be parity dependent, but not dependent on the overall transmission intensity. Can these factors be explored in a multivariate linear regression of the two cohorts and pregnancy/afucosylation levels? If the afucosylation is dependent on overall transmission AND parity, what mechanism may underpin this?

*VAR2CSA expressing parasites are indeed solely found in pregnant women, which is why parity can be used as proxy for exposure. However, exposure to VAR2CSA still requires infection and multiple factors of course determines whether a pregnant woman becomes infected with *P. falciparum*. One of these factors is the level of overall transmission of *P. falciparum* in the given area at a given time, which is why parity can be a more correct estimate of exposure in high transmission areas compared to low transmission areas. In other words: The number of pregnancies can be used as a proxy of possible exposures, and transmission rate estimates the likelihood of a pregnant woman to get infected. To use transmission rate in an analysis would require knowing the transmission rate of the given area during all pregnancies of the subjects included in our analysis, which are data we do not possess.*

Figure 5

This section is a bit clunky, and would suggest an edit. It is quite confusing having CD107a/Ab level and Fc fucosylation switched on the axis A/B – would suggest modify for ease of reading. My interpretation of the data is that – 1) IgG levels are the same between low/high fucosylation. 2) Level of fucosylation strongly associated with degranulation (but total IgG level is not). If this is the main point authors trying to make, I suggest an edit of text/figure.

Figure 5 and figure legend were modified as suggested.

Line 244 – levels of VAR2CSA IgG between groups – can you include a statistical analysis? And also is this total IgG? A comparison of IgG1 would be more informative.

The comparison is for VAR2CSA-specific IgG as indicated. It is correct that a comparison of IgG1 levels would be more informative. However, IgG1 is by far the most abundant isotype of VAR2CSA-specific IgG. Statistics were also added to the sentence (lines 280-5 in "Show all markup" view)

"The samples had similar VAR2CSA-specific IgG levels ($p = 0.80$, Mann-Whitney test; Fig. 5B). However, only VAR2CSA-specific IgG from individuals with low VAR2CSA-specific Fc fucosylation caused marked NK-cell degranulation-induced expression of CD107a ($p = 0.0003$, Mann-Whitney test; Fig. 5C)".

Line 252 – Is there any evidence that the detection of PAM1.4 in ELISA can be impacted by secondary/detection antibodies be impacted by afucosylation levels?

So far we have not encountered any change of affinity by secondary antibodies towards any glycoengineered mAbs (e.g. ref 35). This is in line with the fact that the Fc-glycan is located in the inside of the Fc-domain, which is not accessible by regular antibodies.

Do any of the other antibody characteristics (galactosylation/sialylation/bisection GlcNAc – as in Supplementary Figure) also associated/correlate with NK cell degranulation? Are these characteristics all associated with afucosylation levels?

We have addressed this point by including the following sentence (lines 290-3 in "Show all markup" view):

"Apart from fucosylation, only anti-VAR2CSA IgG1-Fc galactosylation correlated with NK cell degranulation ($r = 0.44$, $P = 0.05$). However, this was only borderline statistically significant, reaffirming previous observations of slightly increased Fc γ R1111 affinity of afucosylated IgG with increased Fc galactosylation"

In the subset of samples tested for NK cell degranulation, the percentage of anti-VAR2CSA IgG1 fucosylation was negatively correlated with galactosylation ($r = -0.45$, $p = 0.04$) but not with sialylation ($r = -0.13$, $p = 0.57$) or bisection GlcNAc ($r = 0.25$, $p = 0.28$).

Discussion

There is a lot of discussion in the results sections that could be moved to discussion to make a longer discussion/conclusion

For this manuscript we chose to go for a combined Results and discussion, allowing for a dynamic discussion as the data are explained, but fully acknowledge that this matter of preference. Please also see our response to a minor comment by Reviewer 1.

Methods queries

Is the purity of the antigen specific IgG used for mass spec known?

The methodologies applied here has been more extensively verified in our previous works. Antigen-specific antibody depletion for sera, and enrichment in the antigen-absorbed-eluted fraction is generally a good indicator for specific absorption. This can sometimes be clearly seen by completely different IgG-glycosylation pattern comparing total IgG to antigens-specific IgG. Negative controls are always taken along (seronegative samples). The mass spec intensities of the antigen-specific IgG1 glycopeptides were at least higher than the mean intensities plus 10xstandard deviation of Dutch seronegative control samples ($n = 5$). This has now been explained better in the Method section (386-

8 in "Show all markup" view). In addition, a clear skewing of the intensities of the different glycoforms was evident compared to IgG total, strongly suggesting a strong enrichment of antigen-specific IgG.

Reviewer comments, second round of review: - -

Reviewer #1 (Remarks to the Author):

The authors have responded to my requests and those of other reviewers and present an improved version of their manuscript. I recommend acceptance.

Reviewer #2 (Remarks to the Author):

The reviewer is happy with the revised manuscript and suggestions have been incorporated

Reviewer #3 (Remarks to the Author):

All comments have been addressed. Congratulations to authors for interesting and important study.